A cross-sectional study of the nasal and fecal microbiota of sows from different health status within six commercial swine farms

Arruda Andreia G. arruda.13@osu.edu 1
Deblais Loic 1
Hale Vanessa L. 1
Madden Christopher 1
Pairis-Garcia Monique 2
Srivastava Vishal 1
Kathayat Dipak 1
Kumar Anand 3
Rajashekara Gireesh 1
1 Department of Veterinary Preventive Medicine, The Ohio State University , Columbus , OH , United States of America
2 Department of Population Health and Pathobiology, North Carolina State University , Raleigh , NC , United States of America
3 Biosecurity and Public Health Group, Bioscience Division, Los Alamos National Laboratory , Los Alamos , NM , United States of America
Kant Ravi
Electronic publication date: 2021 Sep 17
Publication date: 2021
Volume: 9
Electronic Location ID: e12120
Received 2020 Nov 5; Accepted 2021 Aug 16
Copyright: ©2021 Arruda et al.
Copyright year: 2021
Copyright holder: Arruda et al.
License: This is an open access article distributed under the terms of the Creative Commons Attribution License, which permits unrestricted use, distribution, reproduction and adaptation in any medium and for any purpose provided that it is properly attributed. For attribution, the original author(s), title, publication source (PeerJ) and either DOI or URL of the article must be cited.
License URL: https://creativecommons.org/licenses/by/4.0/

Keywords: Swine, Swine microbiota, Nasal and fecal microbiota, Cull sows, Swine health

Funding: Los Alamos National Laboratory (LANL) internal Grant 20170671PRD2 This project was supported by a Los Alamos National Laboratory (LANL) internal Grant (20170671PRD2). The funders had no role in study design, data collection and analysis, decision to publish, or preparation of the manuscript.

==============================
Background

Cull sows are a unique population on swine farms, often representing poor producing or compromised animals, and even though recent studies have reported that the microbiome is associated with susceptibility to diseases, the microbiome of the cull sow population has not been explored. The main objective of this study was to investigate whether there were differences in fecal and upper respiratory tract microbiota composition for groups of sows of different health status (healthy, cull, and compromised/ clinical sows) and from different farms (1 to 6).

Methods

Six swine farms were visited once. Thirty individual fecal samples and nasal swabs were obtained at each farm and pooled by five across health status and farm. Samples underwent 16S rRNA gene amplicon sequencing and nasal and fecal microbiota were analyzed using QIIME2 v.2021.4.

Results

Overall, the diversity of the nasal microbiota was lower than the fecal microbiota (p < 0.01). No significant differences were found in fecal or nasal alpha diversity by sow’s health status or by farm. There were significant differences in nasal microbial composition by farm and health status (PERMANOVA, p < 0.05), and in fecal microbiota by farm (PERMANOVA, p < 0.05), but not by health status. Lastly, at the L7 level, there was one differentially abundant taxa across farms for each nasal and fecal pooled samples.

Discussion

This study provided baseline information for nasal and fecal microbiota of sows under field conditions, and results suggest that farm of origin can affect microbial diversity and composition. Furthermore, sow’s health status may have an impact on the nasal microbiota composition.

Introduction

Culling refers to the process of removing animals from a breeding herd in order to optimize productivity and profitability. This is an essential and common practice conducted on commercial swine herds and it is typical for large systems to cull up to 50% of the herd on an annual basis (D’Allaire, Stein & Leman, 1987; Stein et al., 1990; Stalder et al., 2003). Sows, which refer to reproductively mature females, are normally culled for reasons that include but are not limited to low production efficiency, age, poor reproductive traits, and/ or presumed compromised immune status. However, the definitive underlying reason for culling is often not completely elucidated or recorded. In addition, culls sows may remain on the farm, mixed with the general population, for a significant time period as transportation of mature pigs can be logistically challenging. This results in the maintenance of a consistent subpopulation of potentially compromised animals that may serve as a source of pathogens to the general population of healthy sows. Even though recent work has focused on the role of cull sows on disease transmission outside of the farm (i.e., auctions, sale yards etc.) (Sutherland, 2018), little to no work has been devoted to assessing the role of cull sows in the maintenance and re-emergence of pathogens and disease on-farm.

The microbiome is defined as the microbial community within or on a body site and is known to play an important role in human and animal health (Round & Mazmanian, 2009; Young, 2017; Nowland et al., 2019) and animal production (Jami, White & Mizrahi, 2014). The study of the swine microbiome and its impact in respiratory and systemic diseases has been an emerging area of study within the past few years (Niederwerder, 2017). More specifically, published papers have focused on investigating the associations between the piglet gut or fecal microbiome and intestinal immunity development (Schokker et al., 2014), and performance upon the presence of pathogenic viruses and their co-infections (Niederwerder et al., 2016; Ober et al., 2017). A recent paper has reviewed several publications on piglets’ intestinal microbiome and its association with post weaning diarrhea, with a focus on antimicrobial use (Gresse et al., 2017). A few other studies have also described the associations between the piglet nasal microbiota and presence of pathogenic bacteria such as Staphylococcus sp and Haemophilus parasusis (Weese et al., 2014; Espinosa-Gongora et al., 2016).

Previous work has explored the impact of health status and geographic location on swine microbiota (Correa-Fiz, Fraile & Aragon, 2016); in addition, recent work conducted under laboratory and semi-experimental conditions has determined that the gut and nasal microbiota are associated with susceptibility to respiratory diseases such as porcine respiratory and reproductive syndrome (PRRS) (Niederwerder et al., 2016), porcine circovirus type 2 associated disease (Niederwerder et al., 2016), and Glasser’s disease (Correa-Fiz, Fraile & Aragon, 2016). However, the vast majority of the studies published in the literature focused on piglet populations and did not investigate the effect of health status or farm of origin on the microbiota of sows.

Therefore, the objectives of this study were to:

(1) Characterize baseline fecal and nasal microbiota of adult sows under field conditions,

(2) Investigate the association between health status (including healthy, cull, and compromised/ clinical sows) and nasal and fecal microbial composition and diversity. For the purpose of this study, cull sows were defined as adult females that were identified to be removed from the herd; and compromised or clinical sows were defined as sows presenting acute clinical signs of disease that could be visually assessed by the attending veterinarian or by farm personnel.

(3) Assess the association between farm of origin and nasal and fecal microbial composition and diversity.

Materials & Methods

Study population and sampling protocol

This study has been approved by The Ohio State University’s Institutional Animal Care and Use Committee (IACUC Protocol #2017A00000060), as well as by the Institutional Biosafety Committee (IBC Protocol #2017R00000041).

A cross-sectional observational study was conducted during July of 2017. The project included enrollment of sows distributed within six privately owned commercial farrow-to-wean swine farms from the same production company located in the US Mideast region. A commercial farm was defined as a swine farm that raised pigs with the intent to commercialize them (in this case of farrow-to-wean farms, the farms were producing and selling weaned piglets to downstream farms that would raise them to market). All sows were fed commercial diets from the same source that met the Nutrient Requirements of Swine (National Research Council, 2012). Basic farm descriptors are shown in Table 1. Healthy sows were defined as adult females with no evidence of clinical signs of disease at the time of data collection. Healthy sows were included regardless of stage of production (i.e., pregnant, lactating or weaned). Cull sows were defined as adult females that were identified to be removed from the herd based on the standard operating procedure implemented by the farm staff. Investigators were using sow cards (paper records) in order to identify culled sows. The primary reason for animals to be culled was if they did not achieve acceptable reproductive performance. Compromised or clinical sows (the two terms are used interchangeably throughout the manuscript) were defined as sows presenting acute clinical signs of disease that could be visually assessed by the attending veterinarian or farm staff. This included the following categories: lameness, severe shoulder ulcers, active abortion, and poor body condition score. Each farm contributed an equal number of sampled sows for each health status examined herein (healthy, cull and compromised / clinical); with 10 sows sampled for each health status in each farm.

Table 1 Farm demographics and general information for the source population of the study.

Descriptor	Farm 1	Farm 2	Farm 3	Farm 4	Farm 5	Farm 6	
Date sampled	6/26/2017	7/10/2017	7/10/2017	7/12/2017	7/12/2017	7/18/2017	
Number of sows	1,300	2,700	1,100	2,400	5,500	4,000	
Cull sow housing type	Crates	Crates	Crates	Crates/Pens	Crates/Pens	Crates	
Cull sow removal frequency	Every 2 weeks	Every 1–2 weeks	Every 2 weeks	Every 2 weeks	Weekly	Every 2 weeks	
Mean (SD) parity	3.4 (1.6)	3.5 (2.2)	3.3 (1.8)	3.9 (2.2)	4.3 (3.0)	3.4 (1.8)	
Top three listed reasons for ‘compromised/ clinical’ sows (%)	Low body condition (50), abortion/ discharge (40), lameness (10)	Low body condition (40), abortion/ discharge (30), shoulder ulcer, abcess, other/ missing (10 for each)	Low body condition (40), shoulder ulcer (20), other/ missing (20)	Low body condition (50), abortion/ discharge (30), lameness and shoulder ulcer (10 each)	Low body condition (70), abortion/ discharge (20), shoulder ulcer (10)	Low body condition and shoulder ulcer (40 each), abortion/ discharge (20)	
Top three listed reasons for ‘cull’ sows (%)	Low performance (70), return to estrus (30)	Low performance (40), return to estrus (30), missing information (30)	High parity (40), low performance (30), missing information (30)	Low performance (100)	Missing information (100)	Low performance (80), missing information (20)	
Notes.

Low performance included notes that had any of the following parameters: low born alive, low number of weaned pigs, low milk production, and low performance.

Farms were visited once during the month of July of 2017 for collection of fecal samples and nasal swabs from all sows within a farm on the same day. Fecal samples were used in this study as a non-invasive way to examine the gut microbiota. A total of 30 fecal samples and 30 nasal swabs were obtained per farm (ten for each type of animal) for a total of 360 samples (180 fecal samples and 180 nasal swabs). A schematic of sampling design is presented in Fig. 1, along with relative geographical location for the sampled farms.

Figure 1 Sampling protocol with summarized sample processing steps (A) and relative locations (B) of the six participating farms.

Numbers represent farms 1 to 6. Actual map is not shown for confidentiality reasons.

For nasal swab sampling, pigs were restrained using a snare and the swab (Puritan™ PurFlock™ Ultra Sterile Flocked Swabs) was inserted into the ventral passage of each nostril. The nasal swab was kept in the cavity for three seconds (per nasal passage) and placed in a dry tube. Following this, a fecal sample was collected using digital palpation of the rectum to remove feces with a sterile glove. Fecal samples were then placed into sterile bags.

Sample pooling, genomic DNA extraction and sequencing

Fecal and nasal swabs were collected and placed in tubes on dry ice immediately after collection. Samples were then transported to the Food Animal Health Research Program (The Ohio State University) laboratory in Wooster, Ohio, and transferred into a −80 °C freezer within 24 h, where they remained until extraction. For nasal swabs, two milliliters of phosphate-buffered saline (PBS) were added to each nasal swab tube, tubes were vortexed for 30 s. The samples were divided into 2 subsamples: one 1 ml sample was stored at −80 °C for future use and the other 1 ml sample was centrifuged at 13,000 g for 15 min. The resulting pellet was resuspended in 250 µl of PBS and used for DNA extraction. Fecal and nasal swab samples were pooled across 5 animals of the same status by farm; so each farm contributed two pools of each health status containing samples of five animals each, as shown in Fig. 1. The samples selected to be part of the same pool were randomly selected, and the nasal and fecal pool sets were composed of the same animals. The pools consisted of 0.5 ± 0.015 g of feces per sample and 45 µl of nasal swab suspension per sample. There were 72 pooled samples in total (36 fecal and 36 nasal swab pooled samples) and the pooled sample was considered our unit of analysis. Genomic DNA extraction of the pooled fecal (0.5 ± 0.015 g) or nasal (225 µl) swab samples was conducted using the PureLink Microbiome DNA Purification Kit (Life Technologies, Invitrogen Corp.) followed by RNAse treatment (10 units for 1 h), as previously described (Deblais et al., 2018). Four blank (empty) extractions were also performed as negative controls and sequenced along with the rest of the samples. It is important to note that this commercial kit lyses the cells using chemical (lysis buffer + enhancer [guanidine isothiocyanate]), thermic (65 °C for 10 min), and mechanical (bead beating at full speed for 10 min) approaches. Next generation sequencing library preparations and Illumina MiSeq sequencing were conducted at the Bioscience Division, Los Alamos National Laboratory, NM.

Briefly, DNA samples were quantified using a Qubit 2.0 Fluorometer (Invitrogen, Carlsbad, CA) and DNA quality was checked on a 0.6% agarose gel using E-gel electrophoresis system (ThermoFischer Scientific, Grand Island, NY). A total of 50 ng DNA was used to generate amplicons of the V3–V4 hypervariable regions of the bacterial and Archaeal 16S rRNA gene using primers 341F and 806R. The first round of PCR amplified the V3-V4 region using KAPA HIFI HotStart Ready Mix (Kapa Biosystems, Wilmington, MA) with following PCR conditions; 95 °C for 3 min, 20 cycles of 95 °C for 30 s, 55 °C for 30 s and 72 °C for 30 s and an extension of 72 °C for 5 min. The second round of PCR added the Illumina specific sequencing adapter sequences and unique indexes using the Nextera XT Index Kit v2 (Illumina, San Diego, CA) and KAPA HIFI HotStart Ready Mix (Kapa Biosystems, Wilmington, MA) with following PCR conditions; 95 °C for 3 min, 8 cycles of 95 °C for 30 s, 55 °C for 30 s and 72 °C for 30 s and an extension of 72 °C for 5 min. The amplicons were cleaned up using AMPure XP beads (Beckman Coulter, Indianapolis, IN). The concentration of the amplicons pool was obtained using the Qubit dsDNA HS Assay (ThermoFisher Scientific, Grand Island, NY). The average size of the library was determined by the Agilent High Sensitivity DNA Kit (Agilent, Santa Clara, CA). An accurate library quantification was determined using the Library Quantification Kit –Illumina/Universal Kit (KAPA Biosystems, Wilmington, MA). The amplicon pool was sequenced on the Illumina MiSeq generating paired end 300 bp reads. The amplicon pool was demultiplexed using Illumina’s bcl2fastq. Sequencing data are publicly available at NCBI SRA: PRJNA674564.

Taxonomic microbiota analyses and statistical analyses

Raw, single-end sequence reads were processed using QIIME2 v. 2021.4 (Bolyen et al., 2019). The DADA2 plugin was used to truncate reads at 280 bp and to denoise the single-end reads (Callahan et al., 2016). We opted for single end reads over paired end due to the poor quality of the R2 reads. Taxonomy was assigned in QIIME2 using SILVA version 138.1, with a 99% similarity threshold using the full length 16S rRNA gene classifier (Quast et al., 2013; Yilmaz et al., 2014). All fecal samples with fewer than 5,600 reads and all nasal samples with fewer than 500 reads were removed from analyses. To compare alpha diversity between nasal and fecal samples, all samples were rarefied at 500 reads. We also rarefied at 500 reads to compare nasal samples by farm and health status. To compare fecal samples by farm and health status, we rarified at 5,600 reads. In total, we retained all fecal samples (n = 36) and 19 out of 36 nasal samples for further analysis. To assess the effects of rarefaction, we also analyzed fecal and nasal samples without rarefaction (excluding fecal samples with fewer than 5,600 reads and nasal samples with fewer than 500 reads). We also analyzed nasal samples (rarefied and non-rarefied) excluding samples with fewer than 1,000 reads.

Alpha diversity was analyzed using Shannon and Faith’s Phylogenetic Diversity (PD) indices. A Kruskal–Wallis test was used to compare alpha diversity and read numbers by health status and farm. An adjusted P-value lower than 0.05 was considered significant.

Beta diversity indices were compared using permutational multivariate analysis of variances (PERMANOVA) and permutational analysis of multivariate dispersions (PERMDISP) between Bray Curtis distance matrices. P-values were corrected for multiple comparisons using the Benjamini–Hochberg FDR correction, and values less than 0.05 were considered significant. PERMANOVAs were performed on the fecal and nasal samples by farm and by health status. An analysis of composition of microbes (ANCOM) was used to identify differentially abundant nasal and fecal taxa by farm and health status. An ANCOM performs pairwise comparisons of microbial species ratios and generates a W-value, which indicates how many times a null hypothesis is rejected across comparisons. The maximum W value in any data set is limited by the total number of microbial taxa within that data set. For all samples, we filtered out taxa that had fewer than 10 reads and occurred in fewer than two samples prior to ANCOM analysis. No additional filtering was performed on nasal samples due to the low microbial diversity in the majority of samples. ANCOMs were performed at the L7 level, which is roughly equivalent to a species level; although deeper genome sequencing is necessary for true species differentiation.

Results

One hundred and eighty animals were sampled across the six farms; comprising of a total of 60 healthy sows, 60 cull sows, and 60 compromised / clinical sows. Reasons for sows to be in the ‘compromised’ or ‘clinical’ group included low body condition score (48.3%), acute abortion (23.3%) and shoulder ulcer (15.0%). For the ‘cull’ group, the main reason for culling was low performance (which included low number of weaned pigs, high pre-weaning mortality, and low number of born alive piglets; 50.0%), unknown (missing information on the sow card; 28.3%), and return to estrus (10.0%). A breakdown for most common culling reasons by farm is provided on Table 1. The mean parity for compromised/ clinical animals was 2.9 (SD = 2.0); for cull sows, 5.2 (1.9), and for heathy sows, 3.0 (2.1). A breakdown for parity distribution by farm is provided on Table 1.

16S rRNA gene amplicon sequencing

We obtained a total of 1,759,761 reads with a mean read length of 294 base pairs (Table S1). Of the four negative controls, only one yielded any reads (3 reads total), and all 3 reads were identified as chloroplasts. Chloroplasts, mitochondria, and eukaryotic reads were then removed bioinformatically prior to further analyses. A total of 493 taxa were identified across all samples. The mean number of reads per nasal swab pool (n = 36) was 2,572; the median was 914.5, with a range of 1–27,870 reads (Table S2). Nasal samples with fewer than 500 reads (n = 17) were excluded from analysis. For the 36 pooled fecal samples, we obtained a total of 901,440 reads. The mean number of reads per fecal pooled sample was 25,040, and the median was 22,256, with a range of 5,694–62,677 reads (Table S3). All fecal samples were included in our analyses, which were performed rarefied at 5,600 reads and unrarified. To facilitate a direct comparison between nasal and fecal samples, we also performed a separate analysis with fecal and nasal samples together rarefied at 500 reads and unrarefied.

Alpha diversity results

Overall, nasal samples harbored a significantly lower microbial diversity compared to the fecal samples (Faith’s PD, p < 0.001; Shannon, p < 0.001; Fig. S1A). No significant differences in fecal or nasal diversity (Shannon index) were detected by health status (healthy, cull and compromised/clinical) or by farm (Kruskal–Wallis, all p > 0.05, Fig. 2, Fig. S1B, Table S4).

Figure 2 Alpha diversity by farm and health status.

Microbial diversity (Shannon index) did not differ by farm or health status for pooled fecal samples and pooled nasal swabs. Median and quartiles are shown in the box and whiskers plots. Fecal samples were rarefied at 5600 reads and nasal samples were rarefied at 500 reads.

Beta diversity results

There were significant differences in overall nasal microbial composition by farm (Bray–Curtils PERMANOVA, pseudo-F = 1.83, p = 0.002, Fig. 3A, Fig. S2A, Table S4); although, no pairwise comparisons were significant. Overall nasal microbial composition by health status also differed significantly (PERMANOVA, pseudo-F = 1.64, p = 0.017, Fig. 3B, Fig. S2B) with healthy versus cull as the only significant pairwise comparison (p = 0.033). These differences were not explained by dispersion (PERMDISP by farm: pseudo-F = 0.577, p = 0.795; by health status: pseudo-F = 2.11, p = 0.148) or number of reads by farm or health status (Kruskal–Wallis by farm: p = 0.36; by health status: p = 0.28). When we examined the pooled fecal microbiota of sows, we found significant differences in overall microbial composition by farm (PERMANOVA, pseudo-F = 3.16, p = 0.001, Fig. 4A, Fig. S3A, Table S4) with significant pairwise differences between all farms (p < 0.05) with the exception of farms 4 and 5 (p = 0.08) and farms 5 and 6 (p = 0.1). No significant differences in pooled fecal microbial composition were found based on health status (p = 0.663, Fig. 4B, Fig. S3B).

Figure 3 Nasal microbial communities (Bray Curtis) by (A) farm and (B) health status.

Nasal samples were rarefied to 500 reads and differed significantly by farm (PERMANOVA p = 0.002) and by health status (PERMANOVA p = 0.017). No pairwise comparisons were significant by farm. Healthy versus cull was significant (p = 0.033) by health status.

Figure 4 Fecal microbial communities (Bray Curtis) by (A) farm and (B) health status.

Fecal samples were rarefied to 5600 reads and differed significantly by farm (PERMANOVA p = 0.001) but not by health status (PERMANOVA p = 0.663). Pairwise comparisons were significant between all farms (p < 0.05) with the exception of farms 4 and 5 (p = 0.08) and farms 5 and 6 (p = 0.1). No significant differences were found based on health status (p = 0.663).

In terms of relative abundances, the most abundant phyla in the pooled nasal samples was Proteobacteria, followed by Firmicutes, Actinobacteria and Bacteroidetes (Fig. 5A). The most abundant nasal genera across all pooled samples was Moraxella. At the L7 level, we identified one differentially abundant nasal taxa by farm: a taxa in the Micrococcaeae family, genera Nesterenkonia (ANCOM, W = 32), which was only present on farm 6 (Table S5). No nasal taxa were differentially abundant by health status. In terms of relative abundance in the pooled fecal samples, Firmicutes was the most abundant phyla, followed by Bacteroidetes, Euryarchaeotes, Spirochetes, Actinobacteria and Proteobacteria (Fig. 5B). Within the Firmicutes phyla, the most abundant family was Lachnospiraceae, and the most abundant genera across all fecal samples were Lactobacillus and Methanobrevibacter (an Archaea). At the L7 (roughly species) level in fecal samples, we identified one differentially abundant taxa by farm: it was a taxa in the order Bacteroidales, genera F082 (ANCOM, W = 140), which was broadly present at low abundances across all samples from farms 1, 4, 5 and 6 but almost entirely absent on farms 2 and 3 (Table S6). No fecal taxa were differentially abundant by health status.

Figure 5 Microbial taxa bar plots at the phyla level in (A) nasal samples and (B) fecal samples.

Farm and health status of each sample is identified. All taxa with >1% abundance are included.

Discussion

Cull sows are important from both a disease transmission and welfare perspective but are an understudied subpopulation of animals within commercial swine farms. Studies looking at both fecal and nasal microbiota of adult female pigs are lacking in the literature. Therefore, the main objective of this study was to characterize nasal and fecal microbiota of sows that differed by health status and farm of origin under field conditions.

Our study did not show statistical differences in nasal microbial alpha diversity by health status, or by farm. Few studies have examined swine nasal microbial diversity and reports vary: one study reports significantly decreased diversity linked to Glasser’s disease (Correa-Fiz, Fraile & Aragon, 2016) while another reports no change in diversity linked to methicillin-resistant Staphylococcus aureus (MRSA) carriage (Weese et al., 2014). A few studies have also reported changes in nasal microbiota upon use of antimicrobial treatments, all conducted in young animals (which are different from our population of interest). A study conducted in Spain reported the use of peri-natal treatment with antimicrobials (including penicillin, streptomycin and tulathromycin in one farm, and ceftiofur and tulathromycin in a second farm) in 3–4 week old piglets to be associated with a decrease in nasal microbiota diversity as compared to a period without using these peri-natal treatments (Correa-Fiz et al., 2019). Likewise, a study conducted in the US showed that oxytetracycline administration reduced nasal microbiota diversity in 3-week-old piglets (Mou et al., 2019). In the current study, on-farm antimicrobial treatments were not available to investigators, which did not allow for further investigations. Additionally, the only disease issue reported amongst all six farms in the month prior to sampling was a recent swine influenza outbreak on farm 1. Despite this, we observed no differences in microbial diversity by farm or health status. All farms have reported to not have had outbreaks for the main pathogens of importance in swine including porcine reproductive and respiratory syndrome, Mycoplasma sp. and porcine epidemic diarrhea.

Nasal microbial composition differed both by farm and by health status; and fecal microbial composition differed by farm (all except 4 and 5 and 5 and 6). The differences observed between farms could also not be explained by their geographical location (Fig. 1). Of note, both farm size (number of sows) and parity distribution are higher for farm 5, which could have been contributors for the observed trends in microbiota composition.

In terms of relative abundance in nasal swab samples, Proteobacteria was the largest represented phyla, followed by Firmicutes, Actinobacteria and Bacteroidetes. These results are similar to swine-related nasal microbiota studies conducted in Canada, the United Kingdom and Spain (Weese et al., 2014; Slifierz, Friendship & Weese, 2015; Correa-Fiz, Fraile & Aragon, 2016) which report Proteobacteria as the most abundant phyla, followed by a large abundance in Bacteroidetes, and Firmicutes. In contrast, the abundance of Actinobacteria in our study is higher than what was previously reported (Weese et al., 2014; Correa-Fiz, Fraile & Aragon, 2016), and the abundance of Tenericutes (particularly Mycoplasmacetae family), a phyla which has been recently reclassified into the Firmicutes phylum (Parks et al., 2018), was more prominent in the European study (Correa-Fiz, Fraile & Aragon, 2016) compared to our current study. These differences could be related to animal age given previous studies were conducted with piglets or slaughter animals (approximately six months of age), which are younger than our adult female study population (1–3 years of age). Nesterenkonia was the only taxa identified as differentially abundant by farm, and was only found on farm 6. The relevance of this taxa in swine is unclear; although it has been identified previously in studies on the human lower repiratory tract and gut microbiota (Chander et al., 2017; Li et al., 2019).

Most of the shared taxa for fecal samples in our study were in the Firmicutes and Bacteroidetes families. This agrees with a meta-analysis published in 2017, which reported that those two phyla accounted for nearly 85% of the 16S rRNA gene amplicon sequences among over 930 swine gastrointestinal samples (Holman et al., 2017). This meta-analysis also pointed out that the main factors influencing swine gut microbiota were study itself, including technical factors such as extraction and sequencing, gastrointestinal sample location, and animal age, which makes comparisons between studies a challenge. A study by Kim & Isaacson (2015) analyzed fecal samples of various aged pigs (including sows) and reported that the most common microbial families in sows were Prevotellaceae, Ruminococcaceae, Lachnospiraceae, and Streptococcaceae, which were also present across most of the fecal samples in our study. Those authors also reported a large proportion of “unclassified” taxa, which did not occur in our case.

Even though the farms enrolled in the current study were under the same management practice (i.e., herd veterinarian, health and biosecurity protocols, treatment protocols, feed source), there were likely inherent differences regarding past disease challenges, herd productivity, and physical characteristics of the farm (e.g., farm size, number of workers, environment, biosecurity etc.) that were not fully captured or available to investigators in this study, and may have influenced differences observed. For example, farm size and parity distribution were among some of the differences observed between farms that were different in regards to microbial composition metrics (Table 1).

An important limitation of this study includes the pooling of samples (five sows were represented in each pooled sample). This resulted in only two samples per health status per farm, a small sample size that may have limited our statistical power. This pooling further limited our ability to evaluate the relationship between individual-level variables such as parity, age, production stage and reproductive performance on nasal and fecal microbiota. A second limitation includes the lack of clean/blank swabs extracted as negative controls. Low biomass samples, such as some of the nasal samples in this study, are highly subject to contamination, and we cannot rule out the possibility of swab contaminants in these samples. However, we did include negative controls for extraction (kits) and sequencing, and we analyzed the nasal sample data with and without rarefaction and after removing samples with fewer than 500 or 1,000 reads in an attempt to eliminate the samples at greatest risk of contamination. The low number of reads in the nasal samples likely further limited our statistical power and ability to distinguish true differences between groups. Moreover, in the nasal sample analysis, rarefaction at 500 reads is a low sampling depth that could have limited our ability to detect and compare the true level of diversity within these samples. Despite this, when we analyzed nasal samples with and without rarefaction at a minimum threshold of either 500 or 1000 reads, the results were comparable in all 4 analyses (Table S4). With all these considerations, the results presented herein should be considered preliminary, and interpreted as an initial baseline for our population (s) of interest.

In addition, we further acknowledge the potential for misclassification bias when selecting study subjects. This stems from the fact that on swine farms, the use of “sow cards”, which are paper-based animal records, is a common way of keeping production records and “flagging” animals that need to be culled. In many cases, the reason for culling is not specified. As such, during farm visits, investigators relied on these cards to identify animals that were to be culled, which were then included in the study. Confirmation with the farm manager or farm personnel was attempted whenever possible to minimize this potential source of bias. Future studies could expand sample size and should focus on exploring the association between the fecal and nasal microbiota and detailed demographic (e.g., age), health (e.g., antimicrobial treatment history), and production parameters at the animal or herd levels.

Conclusions

In conclusion, this study provided baseline information for nasal and fecal microbiota of sows under field conditions. Our results suggest that farm of origin can affect nasal and fecal microbial diversity and composition, and health status of animals (compromised/clinical, healthy and cull) can affect alpha diversity. Given the study limitations, these analyses should be expanded to a larger number of animals in the future.

Supplemental Information

Supplemental Information 1 Number of reads in each sample (Chloroplasts, mitochondria, and eukaryotic reads have been removed)

Click here for additional data file.

Supplemental Information 2 Nasal samples - Average number of reads by farm and health status (samples with fewer than 500 reads were excluded)

Click here for additional data file.

Supplemental Information 3 Fecal samples - Average number of reads by Farm and Health Status (samples with fewer than 5600 reads were excluded)

Click here for additional data file.

Supplemental Information 4 Microbial community analysis of nasal samples with and without rarefaction at 500 reads and 1000 reads. Microbial community analysis of fecal samples with and without rarefaction at 5600 reads

Click here for additional data file.

Supplemental Information 5 L7 feature table for pooled nasal swab samples

Click here for additional data file.

Supplemental Information 6 L7 feature table for pooled fecal swab samples

Click here for additional data file.

Supplemental Information 7 Alpha diversity analysis (Faith PD index) for pooled fecal and nasal swabs

Microbial diversity by a) sample type and b) farm and health status. In A, all nasal and fecal samples were rarefied at 500 reads. In B, fecal samples were rarefied at 5600 reads and nasal samples were rarefied at 500 reads. Median and quartiles are shown in the box and whiskers plots. N = 36 fecal and 19 nasal swabs. *p <0.001

Click here for additional data file.

Supplemental Information 8 Nasal microbial communities (Jacaard) by a) farm and b) health status

Click here for additional data file.

Supplemental Information 9 Fecal microbial communities (Jacaard) by a) farm and b) health status

Click here for additional data file.

The authors would like to acknowledge farm owners, managers and personnel, and undergraduate students for help with sampling collection.

Additional Information and Declarations

Competing Interests

Author Contributions

Field Study Permissions

Data Availability

The authors declare there are no competing interests.

Andreia G. Arruda conceived and designed the experiments, performed the experiments, prepared figures and/or tables, authored or reviewed drafts of the paper, and approved the final draft.

Loic Deblais performed the experiments, analyzed the data, prepared figures and/or tables, authored or reviewed drafts of the paper, and approved the final draft.

Vanessa L. Hale, Christopher Madden and Anand Kumar analyzed the data, prepared figures and/or tables, authored or reviewed drafts of the paper, and approved the final draft.

Monique Pairis-Garcia conceived and designed the experiments, performed the experiments, authored or reviewed drafts of the paper, and approved the final draft.

Vishal Srivastava and Dipak Kathayat performed the experiments, analyzed the data, authored or reviewed drafts of the paper, and approved the final draft.

Gireesh Rajashekara conceived and designed the experiments, prepared figures and/or tables, authored or reviewed drafts of the paper, and approved the final draft.

The following information was supplied relating to field study approvals (i.e., approving body and any reference numbers):

This study has been approved by The Ohio State University’s Institutional Animal Care and Use Committee (Protocol #2017A00000060), as well as by the Institutional Biosafety Committee (Protocol #2017R00000041).

The following information was supplied regarding data availability:

The sequences are available at NCBI: PRJNA674564.

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
