# Peer review of "A cross-sectional study of the nasal and fecal microbiota of sows from different health status within six commercial swine farms"

_PeerJ, doi:10.7717/peerj.12120_

## Round 0.1 · original submission · Major Revisions

Please address all the points raised by both the reviewers for the manuscript.

·

Basic reporting

The language is mostly clear, just some small editing required:
- Line 27: check sentence structure
- 30: clarify: each sample from an individual animal?
- 32 and elsewhere: 16S rRNA _gene amplicon_ sequencing (you didn't sequence RNA!)
- 130: /hr = ?
- 163: "separately" = unclear
- 166: diveristy = typo
- 182 and elsewhere: 0-1 decimals is sufficient and more adequate
- 196: belongs rathet to the next chapter?
- Fig 3 caption: the caption says FECAL microbiota in 3b, is this a typo?
- 222: very unclear. Do you mean individual OTUs or orders or what?
- 223: plyLum = typo
- 226 and elsewhere: please use tables rather than long lists of taxa.
- 266: unclear sentence, rewrite
- 321: was this really at the SPECIES level?
- 341: explain more clearly!

Tables:
- Table 3: this does NOT look like relative abundance data. What is the unit? Absolute counts can't be compared like this.
- Report ALL observed families / genera as a supplementary table.
- Report the parities etc. PER FARM (table!).

Figures:
- Fig 2: show P values in the fig.
- Fig 4: legends too small!
- Fig 4a: why are there several missing pools here?

Data:
- The data provided as attachment is NOT the raw sequencing data! The BIOM taxonomy table is also nice to have, though.

Experimental design

This is original primary research with a relevant and well defined research question. However, there are some major problematic aspects in the experimental design and several missing data which should be reported.

- Negative controls are not reported at all. The nasal mucosa is a low-abundance microbiota, and proper negative controls are absolutely essential to ensure validity of results. Please perform DNA extraction and sequencing from unused sampling instruments. This of course should have been done together with actual samples, but better late than never.
- The pooling of the samples is unfortunate, as it drastically reduces the statistical power and resolution of the data. Considering the low costs of 16S sequencing currently, I wonder if this is really absolutely necessary. Could you consider re-doing the sequencing individually?
- Some potentially significant background information is missing. Report details of feeding, and of culling protocols in each farm.
- The production stage of the healthy sows very likely strongly influences the microbiota. This must be more adequately considered (another reason to perform sequencing individually).
- Report how and when the samples were frozen.
- Report if the PureLink protocol includes mechanical lysis. This is critical.
- An old version of QIIME and an EXTREMELY old version of the Greengenes database were used. The bioinformatics need to be re-run using up-to-date version (preferably something else than Greengenes).
- Report read counts per sample type. Apparently these were very small per sample.
- Why was OTU based cutoff used, rather than read count based? The nasal microbiota samples could very well really be low diversity. This kind of cut-off may bias the results.
- Fig 3: the clustering doesn't seem very clear. Perform PERMDISP to check if it's actually just about dispersion differences.
- 269: Can't compare absolute read counts! Use relative abundances.
- 283: Impossible to compare OTU counts between studies! Alpha diversities perhaps a bit better, but should probably re-analyze the sequencing data from beginning.
- 295: Please analyze WHAT was were the differences revealed by PCoA.
- 316 and elsewhere: should also consider effects of feed!
- 325: Family level too broad for any meaningful comments.

Validity of the findings

This study provides useful baseline data on sow microbiotas, if the shortcomings are corrected.

Data: please see part 2.
- Negative controls are missing and the pooling drastically weakens the study.
- Much of essential metadata is missing.
- The sequencing data is not provided.

Conclusions: these are well in line with the reported results and not exaggerated or overly speculative. After complementing the study and reporting as suggested here, the authors may be able to interpret the results a bit further.

Additional comments

Studies like this are important and very welcome. I do hope that you are able to improve the work as suggested and would like to see the work published in the improved form.

Reviewer 2 ·

Basic reporting

The manuscript describes the fecal and nasal microbiota composition on a particular group of sows: the ones marked to be removed from the herd in the typical common practice in large farms. Because cull sows remain on the farm, mixed with general population, this analysis is interesting to assess the role of cull sows in the maintenance and re-emergence of pathogens and disease in the farms. The basic idea underneath this study could be evaluated through the microbiota samples taken, however the analysis is out-of-date (qiime 1 is no longer supported since 1/1/18) and there are many concerns that need to be resolved before going public.
The use of English is correct throughout the manuscript.
The raw data is not provided.
The introduction needs more detail and references to state the problem. I suggest to improve the description on previous knowledge on microbiota and diseases in swine. Some examples are:
Espinosa-Gongora, C., Larsen, N., Schønning, K., Fredholm, M. & Guardabassi, L. Differential Analysis of the Nasal Microbiome of Pig Carriers or Non-Carriers of Staphylococcus aureus. PLoS One11 (2016)
Schokker D, et al. Early-life environmental variation affects intestinal microbiota and immune development in new-born piglets. PloS One. 2014;9:e100040. doi: 10.1371/journal.pone.0100040.
Ober RA, et al. Increased microbiome diversity at the time of infection is associated with improved growth rates of pigs after co-infection with porcine reproductive and respiratory syndrome virus (PRRSV) and porcine circovirus type 2 (PCV2) Vet. Microbiol. 2017;208:203–211. doi: 10.1016/j.vetmic.2017.06.023
Gresse R, et al. Gut microbiota dysbiosis in postweaning piglets: Understanding the Keys to Health. Trends Microbiol. 2017;25:851–873. doi: 10.1016/j.tim.2017.05.004.
Recently, a study evaluating both nasal and fecal microbiota of piglets in the farm in the association to the microbiota of their sows should be mentioned and discussed:
Correa-Fiz, F., Neila-Ibáñez, C., López-Soria, S. et al. Feed additives for the control of post-weaning Streptococcus suis disease and the effect on the faecal and nasal microbiota. Sci Rep 10, 20354 (2020). https://doi.org/10.1038/s41598-020-77313-6
Please, be precise when citing. Examples:
Line 63. This particular work describes these changes in nasal microbiota, please specify.
Line 67. “…only focused on healthy sow…” this is not correct for ref [10] which describes the microbiota on piglets.
Line 70. I think this is a study more than a “project”.

Experimental design

The M&M section is well-described in general. However, I find several points that can be improved.
Table 1. Also, include the percentage of each disease (lameness, ulcers, abortion) per farm in the case of compromised culls. The total percentage is included in results section but the info per farm can be relevant to see if the distribution is similar.
Line 109. Which kind of swabs were used? Please, specify so that the experiment is reproducible.
Lines 122-1234 Were the pools done randomly across the 5 animals or was there a reason to pool them? Were the same animals included in the different nasal and fecal pools?
Line 160 Why 94% was used. The typical microbiome analysis uses 97 or 99%.
Line 163-164. The statistical analysis on microbiome data considering the two different variables under study are not described.
Line 168. The analysis can be improved including other diversity metrics, typically used in microbiome studies, such as Shannon. This is the typical metrics to measure diversity and the inclusion could largely benefit the manuscript with interesting results.

Validity of the findings

Line 189 Is the total count of 134,609 is for all the samples or mean per sample?
Line 192 if such a low number was used (done-hundred was used as a cutoff) the results presented should be considered preliminary. This low number is not representative of the composition of the nasal microbiota. Moreover, I would like to see as supp material the number of OTUs per sample.
Line 200 Wouldn’t be possible that the absence of difference is due to the low number of reads analyzed? Discuss.
The differences detected by Chao1 index could be due to the way the richness is estimated based on doubletons.
Lines 206-209. This statement is not correct as it is written. The percentage shown in the plot refer to the percentage of variation explained by each axis of the ordination, but this does not mean this is the percentage of variation between groups, as stated. To obtain this percentage (R2), other tests should be applied, such as Adonis function from Vegan package (Okasnen J, et al. Vegan: community ecology package. R package version. 2015;2:3–0). Please, clarify this or perform the correct analysis to add this info.
Lines 221-226 The way this part of the analysis was done, is not included in M&M section.
Line 228. Why do you state significant differences were observed when you previously have shown that “There was no difference in alpha diversity indices…” (line 200) and “Likewise, there was no difference in diversity indices…” (line203)…? Do you mean that the samples clustered by health status? Are you referring exclusively to richness measured by Chao?
Figure 2. Please, show the statistical differences with stars between the correspondent groups.
Line 240. Idem as before (206-226)
Line 267-272 Remember you are always comparing relative abundance, clarify this in the whole parapgraph. Also, I think it’s worth to mention the relative abundance of the differential OTU found instead of the mean number of reads, as it can be more informative.
Table 3. The title is repeated, the second version is correct.
Lines 254-265 The details regarding microbiota composition are not explicitly shown with their relative abundance values, I would suggest to include this information in a bar plot or as supplementary tables with full information.
The supplemental material is not cited throughout the text.
The discussion section needs to be improved.
Line 292. It is true that few studies have examined the swine nasal microbial diversity. So, in my opinion all should be mentioned. The changes in the nasal microbiota has also been evaluated regarding to antibiotic treatments (https://doi.org/10.1016/j.vetmic.2019.108386 and https://www.ncbi.nlm.nih.gov/pmc/articles/PMC6484018/ ) which I think can have an impact in this study too. Is it possible to know the treatments the sows were exposed? If you don’t have this information, at least you should discuss this issue.
There are some facts that are poorly or not at all discussed in the text:
The study was done regardless the stage of the sows, however pregnancy and lactation can change the microbiota composition at least in humans.
The differences found between farms is shown to differ. The clinical status of the farm should be discussed.
The manuscript would have improved a lot if specific pathogens were particularly searched in the microbiome data.
The analysis is limited to phyla mainly, while most of the differences shown in other papers was at family or genus level. Did you analyze at this taxonomy levels? Include the info or discuss the limited findings.
The fact that a different region from 16S rRNA gene (V4-V5) is analyzed in this study compared to literature (V3-V4), can deeply biased the comparisons. This fact is included briefly but not highlighted.

Additional comments

The data obtained in this study can provide relevant information, however it should be improved a lot the way the analysis is presented to be able to get clear insights into this area.

---

## Round 0.2 · Major Revisions

Thanks for addressing the reviewer comments but the manuscript still needs substantial improvement before it can get accepted. Please address all the concerns raised by the reviewer or unfortunately, I will need to reject the manuscript.

·

Basic reporting

Mostly very nice and clear. Just a few, mostly small comments here:
- I couldn't access the raw data by the identifier provided.
- 60: Add year to ref. Sutherland
- 89: Define "commercial" in more detail
- 144: Unclear sentence: WHAT was pelleted and resuspended into what?
- 153: Unclear - the pool is smaller than original samples?
- 192: Clarify the meaning of "filtered"
- 264: Why first part B?
- Fig3: Add captions to a, b ("by farm", "by health status")

Experimental design

I'm happy to see that the data analysis was re-done using current methods. Unfortunately, I'm still afraid that the sample pooling and the partially very small sequencing depths seriously limit the analysis and interpretation of the results.

Using samples with less than 100 reads per pool is highly questionable (I'd be very careful with 1000 reads!). You need to somehow try to validate the analysis results to make sure they are not affected by the very low (and varying) sequencing depths:

- Does the alpha diversity comparison (nasal/fecal) hold if you use rarefaction (or some more sophisticated method) to account for the differences in seq depth? - I'm generally not a fan of rarefaction, but I'm worried when you compare 100 sequences vs 20000 sequences and say that the latter is more diverse.

- Are the PCoAs and other analyses affected by the seq depth differences? I'd generally be very cautious in interpreting the quite weak inter-group differences observed here, especially as the data comes with so many limitations. Also, please check PERMDISP to see whether the reported PERMANOVA results are explained by different dispersions.

- 155: Very nice that you now properly report the use of the negative controls. However, please specify if the blank extractions included empty swabs or were they just empty extractions?
- 169: Why did you use the same number of cycles for both types of samples? The fecal samples are expected to contain much more bacterial DNA (although you quantified the DNA, the nasal swabs likely contain relatively much more host animal DNA).
- 267: Please discuss the reliability of the Nesterenkonia observation (here or in Discussion). Is it realistic to find this genus in this source? Please check the representative sequences for this ASV by BLAST etc.
- 267 & 274: It could be useful to explain the ANCOM results/interpretation a bit, as this is perhaps not the most familiar method for all readers.

Validity of the findings

The manuscript has been significantly improved. I continue to feel that this is an interesting and important topic, but unfortunately the limitations of the data (small N, pooling, and extremely small sequencing depths in the nasal samples) make it challenging to do very many conclusions.

I recommend that the authors consider my remaining critical comments, validate the results of the analyses vs. the limitations of the data, and remove over-optimistic conclusions if they are not really adequately supported by the data.

I think the study still has value as a basic descriptive report for an important field which has not been thoroughly studied. I leave it to the editor to decide whether this type of a report is suitable for PeerJ.

Additional comments

I'm happy to see that the authors have utilized the reviewer comments to significantly improve the manuscript. It's a pity (but understandable at this stage) that it's not possible to add more actual data to the study (individual analyses of animals, etc.). I hope the study will be published here or in another journal, as it does provide useful baseline information for pig research.

---

## Round 0.3 · Major Revisions

Please address all the concerns raised by the reviewer.

·

Basic reporting

Just a small typo on line 210/217: "generates generate".

Experimental design

All other issues have now been addressed, but some major problems remain in the analysis of nasal samples. I didn't previously realize that these were indeed rarefied down to 66 reads. This is really very, VERY small number of reads, and I don't think I've ever seen anybody try to perform a meaningful analysis at such a level (although this problem is not unfamiliar in our own work!). It may still be worthwhile to report these data, but the major limitations need to be still more thoroughly and openly addressed:

- Report clearly the distribution of read depths in nasal samples (at least, medians and standard deviations per experimental group; preferably also for every individual sample e.g. in supplement)
- Depending on the number of samples with adequate read depths (let's say, >1000 or at least >500, really depending on how the analytical results behave), consider omitting even more low-depth samples
- Re-run the analyses without rarefaction, as rarefying to such extremely low level will delete most of your data; compare and discuss the results obtained by either way
- Remove or carefully discuss ALL comparisons possibly affected by this limitation (such as, nasal vs fecal microbiota; also critically evaluate the reliability of nasal microbiota comparisons between experimental groups etc)
- Based on these, discuss the nasal microbiota data still more critically
- Also critically discuss the uncertainty of not analyzing empty swab negative controls vs. the very low read depths in nasal samples

Note: I would be VERY happy to see a 16S qPCR comparison of nasal samples vs. extracts from empty swabs - this would not be a big deal to do, and would considerably contribute to the reliability of the data.

Validity of the findings

See part 2.

Additional comments

Many thanks for considering the reviewer comments and again improving the manuscript. I also highly appreciate providing the revisions in several clear formats to the reviewers, making our job easier.

I apologize for not fully realizing the issue about nasal sample sequencing depths and rarefaction prior to this revision. Based on the PeerJ guidelines, I need to classify this as a major revision, as it may affect interpretation of results, requiring one more round of review.

---

## Round 0.4 · Major Revisions

I am giving a last chance to authors to improve their manuscript as suggested by reviewer 1. If their concerns are not addressed in the next revision I will have to reject this manuscript.

·

Basic reporting

-

Experimental design

Thank you for the additional information on the effects of low read counts and rarefaction. However, I'm afraid I still cannot accept the statistics in the present form. Please re-do all statistics like I already suggested:

- Exclude all samples <500 reads from all analyses. Even this is a VERY low threshold, probably considered much too low for many experts. Below that, it's really not appropriate to perform any meaningful analyses.

- Perform rarefaction to 500 reads in Alpha diversity analyses. It is impossible to reliably compare alpha diversity indices for samples with 500 and 50000 reads (and it's quite obvious that nasal and fecal samples seem different because of this). For other type of analyses, I recommend using the full data (without rarefaction).

- It was very nice for the reviewers to see the effects of various ways of data processing (rarefaction or not, exclusion or not). But it is not very useful to the reader and makes the results more complicated to read. So please report only one version of results, based on an acceptable data analysis (as suggested above).

I still believe a qPCR analysis would nicely complement the results for low-biomass samples, even if the negative control swabs would be from a different lot.

Validity of the findings

see above

Additional comments

I'll be happy to do the full re-review when you provide the properly done statistics.

Reviewer 2 ·

Basic reporting

Comments addressed

Experimental design

Comments addressed

Validity of the findings

Comments addressed

Additional comments

Most of the issues previously raised have been explained and solved out.

Typo in line 240-241 “…with a range of of 1-27,870 reads prior to…”

There is still a sentence that needs to be corrected:

Lines 312-316 when explaining the work of Correa-Fiz et al 2019, the sentence is not correct. IN this study, the use of antimicrobials is compared in the same farms in a period using peri-natal antimicrobial treatment and after, when this treatment was not done at all. So the correct sentence would be:

A study conducted in Spain reported the use of peri-natal treatment with antimicrobials (including penicillin, streptomycin and tulathromycin in one farm, and ceftiofur and tulathromycin in a second farm) in 3-4 week old piglets to be associated with a decrease in nasal microbiota diversity as compared to a period without using these peri-natal treatments.

---

## Round 0.5 · accepted · Accept

Thanks for addressing all the concerns of reviewer 1.

·

Basic reporting

All my concerns have now been sufficiently addressed. Well, some of the texts in the figures are tiny and some of the colors in the figures difficult to distinguish from each other.

Experimental design

All my concerns have now been sufficiently addressed.

Validity of the findings

All my concerns have now been sufficiently addressed.

Additional comments

All my concerns have now been sufficiently addressed.